# Deep Learning-Based Black Spot Identification on Greek Road Networks

**Ioannis Karamanlis** [1], **Alexandros Kokkalis** [1], **Vassilios Profillidis** [1], **George Botzoris** [1], **Chairi Kiourt** [2], **Vasileios Sevetlidis** [2] **and George Pavlidis** [2,*]

[1] Department of Civil Engineering, University Campus at Kimmeria, Democritus University of Thrace, GR-67100 Xanthi, Greece; iokarama@civil.duth.gr (I.K.); akokkal@civil.duth.gr (A.K.); vprofill@civil.duth.gr (V.P.); gbotzori@civil.duth.gr (G.B.)

[2] Athena Research Center, University Campus at Kimmeria, GR-67100 Xanthi, Greece; chairiq@athenarc.gr (C.K.); vasiseve@athenarc.gr (V.S.)

\* Correspondence: gpavlid@athenarc.gr

**Abstract:** Black spot identification, a spatiotemporal phenomenon, involves analysing the geographical location and time-based occurrence of road accidents. Typically, this analysis examines specific locations on road networks during set time periods to pinpoint areas with a higher concentration of accidents, known as black spots. By evaluating these problem areas, researchers can uncover the underlying causes and reasons for increased collision rates, such as road design, traffic volume, driver behaviour, weather, and infrastructure. However, challenges in identifying black spots include limited data availability, data quality, and assessing contributing factors. Additionally, evolving road design, infrastructure, and vehicle safety technology can affect black spot analysis and determination. This study focused on traffic accidents in Greek road networks to recognize black spots, utilizing data from police and government-issued car crash reports. The study produced a publicly available dataset called Black Spots of North Greece (BSNG) and a highly accurate identification method.

**Keywords:** AI; black spot; road safety; dataset



## 1. Introduction

Road safety is a crucial issue that affects not only the individuals involved in road accidents, but also society as a whole. The cost of road accidents in terms of human lives lost, physical and emotional suffering, and financial losses is enormous. Thus, it is important to understand the factors that contribute to road accidents and to develop strategies to reduce the number and severity of these incidents.

One of the most important steps in this process is the identification of "black spots", areas where the number of accidents is significantly higher compared to other parts of the road network. The identification of black spots is crucial for prioritizing road safety interventions and evaluating their effectiveness in reducing the number of accidents. These events can range from minor incidents, such as fender benders, to serious crashes, resulting in fatalities or severe injuries. Thus, identifying these areas provides insights into the underlying causes of these accidents.

For example, black spot analysis can reveal the presence of road design or infrastructure issues that may contribute to accidents, such as poor lighting, confusing road signs, and a lack of pedestrian crossings. By understanding the factors that contribute to accidents in black spot areas, authorities can develop strategies to reduce the frequency

and severity of accidents in these locations, such as adding additional lighting, improving signage, or installing pedestrian crossings, to reduce the risk of accidents [1].

In addition, methodological analysis can also help identify driver behaviour issues that may contribute to accidents, such as speeding, distracted driving, or reckless behaviour. By targeting these behaviours through enforcement, education, and engineering, authorities can reduce the frequency of accidents in black spot areas. However, the identification of black spots can be challenging for several reasons.

Moreover, it is difficult to determine the exact number of accidents that occur in black spots compared to regular road segments, as this can vary greatly depending on the location and road network. However, it is generally accepted that black spots, or areas with a higher concentration of road accidents, experience a disproportionate number of crashes compared to regular road segments [2]. Therefore, identifying black spots requires a multidisciplinary and comprehensive approach that considers a variety of data sources and analytical techniques, and the utilization of machine learning. In particular, deep learning has recently gained popularity in transportation research, despite its potential, its use in the identification of black spots has not been widely explored.

*Contribution*

Various methods have been used for black spot prediction, including traditional statistical analysis, machine learning, and deep learning methods. These methods vary in their level of complexity, data requirements, and accuracy, and can be used to predict road accidents based on a variety of factors, such as road design and infrastructure, driver behaviour, and weather conditions. The goal of this paper is to provide an overview of the different black spot prediction methods, their strengths and limitations, and to discuss their potential applications in the field of road safety. The contribution of this work is threefold as it offers an in depth literature review, it publishes a novel dataset on black spot identification (BSNG), and it establishes an identification benchmark of these problematic areas on Greek road networks.

## 2. Literature Review

### 2.1. Terminology

Black spot identification is a topic of ongoing research and has been the focus of numerous studies in recent years. This is due to the increasing number of road accidents and the need to improve road safety. The objective of black spot identification is to identify locations on the road network where accidents are occurring more frequently, with the aim of reducing the frequency and severity of accidents in these areas.

A "black spot" in road networks refers to a specific location or stretch of road that has a high frequency of accidents or incidents, that often result in fatalities or serious injuries. These areas can be identified by analysing data from police reports, traffic accidents, and other sources. The goal is to address the underlying issues that contribute to these accidents and to make these areas safer for all road users.

The term itself was first used in the early 1760s [3] and 1770s [4,5]. It was popularized by road safety researchers and practitioners in Australia [6,7] and New Zealand [8,9], who used it to describe areas where a disproportionate number of accidents occurred. The term was later adopted by road safety authorities and organizations worldwide and has since become a widely used term in the field of road safety.

The choice of the words "black" and "spot" highlights the importance of identifying these areas and has helped bring attention to the need for improvements to road design, infrastructure, and driver behaviour to reduce the frequency of accidents in these locations.

Different countries may have different definitions for what constitutes a black spot in their road networks, as shown in Table 1. The exact criteria used to identify black spots can vary depending on the country, type of road network, and available data [10]. For example, some countries may define a black spot as a location where a specific number of accidents occur over a certain period of time, whereas others may define it based on the severity of

accidents or the number of fatalities. Some countries may also consider other factors such as traffic volume, road geometry, or driver behaviour when identifying black spots [11].

**Table 1.** Different countries have different definitions and criteria for identifying a black spot.

| Country/Area | Methodology | Sliding Window (m) | Threshold | Severity Included | Time Frame (Years) |
|---|---|---|---|---|---|
| Denmark | Poisson | variable length | 4 | No | 5 |
| Croatia | Segment ranking | 300 | 12 | Implicitly | 3 |
| Flanders | Weighted method | 100 | 3 | Yes | 3 |
| Hungary | Accident indexing | 100 (spot)/1000 (segment) | 4 | No | 3 |
| Switzerland | Accident indexing | 100 (spot)/500 (segment) | Statistical, critical values | Implicitly | 2 |
| Germany | Weighted indexing | Likelihood | 4 | No | 5 |
| Portugal | Weighted method | 200 | 5 | Yes | 5 |
| Norwey | Poisson, statistical testing | 100 (spot)/1000 (segment) | 4 | Accident cost | 5 |
| Greece | Absolute count | 1000 | 2 | No | N/A |

Additionally, the methods for collecting and analysing data on road accidents can differ, which can affect the accuracy and reliability of black spot identification. Some countries may use police reports [12,13], whereas others may use more comprehensive data sources, such as road safety monitoring systems or traffic simulation models.

Given these differences, it is important to understand the specific definition of black spots used in a given country to accurately identify and address black spots in the road network. It is also worth noting that the number of black spots in a country's road network can change over time as authorities implement safety improvements and road conditions change. Consequently, it is important to regularly monitor and analyse road safety data to identify and address black spots on an ongoing basis.

Finally, in terms of proportional comparison, it is difficult to determine which country suffers the most black spots in its road network, as this depends solely on the criteria used to define a black spot and the availability and accuracy of the data [14]. It is widely acknowledged that some countries, particularly those with a high traffic volume, are more likely to have a higher number of black spots in their road networks. For example, countries with large populations, high levels of urbanization, or extensive road networks may be more likely to have a higher number of black spots because of the increased risk of accidents in these areas.

*2.2. Methodologies*

2.2.1. Statistical Analysis

The use of statistical analysis techniques, such as regression analysis and time-series analysis, is a common approach for uncovering patterns and relationships between road accidents and factors such as road design, traffic volume, driver behaviour, and weather conditions. These techniques have a long history of use in road safety modelling [15]. Many statistical models have been employed, such as Poisson regression [16,17], binomial regression [18], negative binomial regression [19], Poisson-lognormal regression [20], zero-inflated regression [21], generalized estimation equations [22], negative multinomial models [23], random effects models [19], and random parameter models [24]. Similarly, various models have been proposed for crash severity, including the binary logit, binary probit, Bayesian ordered probit, Bayesian hierarchical binomial logit, generalized ordered logit, log-linear model, multinomial logit, multivariate probit, ordered logit, and ordered probit [25]. Their use in black spot analysis has both advantages and disadvantages.

Statistical analysis provides a systematic and objective way to examine large and complex datasets, thereby reducing the potential for human error and bias in data analysis. These techniques often have the ability to predict future road accidents based on historical data, which can be valuable in identifying black spots and developing countermeasures. However, their use can be complex and requires specialized knowledge, limiting the ability of some organizations to carry out this type of analysis. The accuracy and reliability of the data used in statistical analyses can also be a concern, as inaccurate or incomplete data

can lead to unreliable results. In addition, the lack of contextual information, such as the relationship between road accidents and various other factors, can make it difficult to fully understand the underlying causes of accidents.

### 2.2.2. GIS-Based Analysis

Geographical information system (GIS) technologies can be used to map road accidents and identify hot spots on the road network. GIS technology can uncover relationships between spatial phenomena that are not easily detected using non-spatial databases [26]. Over the past few decades, numerous studies have been conducted on the use of GIS technology in traffic safety and accident analysis, with many organizations and researchers reporting its successful application. These types of analyses include intersection analysis [27], segment analysis [28], cluster analysis [29], and density analysis modelling techniques [29].

However, the GIS technology has shortcomings. It is expensive and requires specialized skills and knowledge. This may limit the ability of some organizations to implement this type of analysis. The quality of the data used in the GIS analysis is critical to the accuracy of the results. If the data are inaccurate or incomplete, the analysis results may also be unreliable. Although GIS technologies are adept at interpreting spatial information, they can provide limited information to understand the temporal factors contributing to accidents, such as driver behaviour. Overall, the use of GIS technology provides a visual representation of the location of black spots on a road network and allows for the integration of multiple data sources.

### 2.2.3. Accident Reconstruction

Accident reconstruction involves the use of various techniques, such as computer simulation models, to recreate the conditions leading up to a road accident in order to understand the causes and contributing factors [30,31]. By simulating the conditions leading to an accident, it is possible to identify the contributing factors and causes of the accident [32]. In addition, these models allow the testing of different scenarios in a controlled environment, which is much safer than testing on real roads. Usually, they can be cost-effective in evaluating the safety of road networks as they eliminate the need for physical testing and save resources [33].

However, this comes with the complexity of developing computer simulation models, which are usually complex and require specialized knowledge and expertise for operation and interpretation. Additionally, they can be limited by the assumptions and simplifications made in the modelling process, which ultimately affects the accuracy of the simulation. Finally, validation of the results can be challenging, and it can be difficult to determine the accuracy and reliability of the results.

### 2.2.4. Road Safety Audits

Road safety audits involve a thorough examination of the road network and the surrounding environment, with a focus on identifying road design and infrastructure issues that may be contributing to accidents [34]. Road safety audits are typically conducted by a team of experts who may use various tools and techniques, such as simulation models, to evaluate the road network [35]. This method is holistic because it considers various aspects of the road network and the surrounding environment, such as road design, traffic volume, weather conditions, and driver behaviour [34].

Holistic approaches aim to understand the overall road scenario and identify areas for improvement. By examining the road network and surrounding environment, it is possible to identify the root causes of accidents, such as poor road design, inadequate signage or lighting, and obstacles that can obstruct the visibility of drivers.

This approach can be time-consuming because it requires experts to visit the road network and perform a comprehensive examination of the surrounding environment. In addition to the workload, the cost of carrying out such a thorough examination of the road network and the surrounding environment can be high, especially if specialized

equipment is needed. In contrast to other analytical methods, the results of the examination may be subjective, as they depend on the skills, experience, and perspective of the personnel conducting it [36].

### 2.3. Deep Learning-Based Methods

In recent years, there has been growing interest in using machine learning, especially deep learning, in transportation research. However, the accuracy of such models depends on various factors, including the type and size of data, the location where the data were collected, and the timing of the predictions. Despite its potential, the application of machine learning for identifying black spots remains limited.

Theofilatos et al. utilized two advanced forms of machine learning and deep neural networks (DNNs) to predict road accidents in real-time. Their study used a dataset that included past accident data and current traffic and weather information from Attica Tollway in Greece [37]. The results showed an accuracy of 68.95%, a precision of 52.1%, a recall of 77%, and an AUC of 64.1%.

Fan et al. applied a classic machine learning method, such as SVM, along with a deep learning algorithm to analyse a large dataset of traffic accident records and various other relevant factors, such as road conditions and weather. They proposed an online FC deep learning model to predict accident black spots. The reported accuracies showed that the deep learning approach outperformed the traditional statistical methods in terms of accuracy, demonstrating the potential for using deep learning in urban traffic accident prediction. The authors stressed the importance of the weather as a predictor [38].

Finally, which method would have been the most effective depends on the specific road network and data available, and often involves a combination of them to provide a comprehensive understanding of the factors contributing to road accidents in black spot areas.

## 3. Creation of the Dataset

In the past, studies have been conducted regarding black spots on Greek road networks. Theofilatos et al. (2012) [39] focused on identifying and analysing the factors affecting the severity of road accidents in Greece, with a specific emphasis on the comparison between accidents that occur inside and outside urban areas. Using road accident data from 2008, two models were developed using a binary logistic regression analysis. The models aimed to estimate the probability of a fatality or severe injury versus a slight injury, as well as the odds ratios for various road accident configurations. The models were tested for goodness-of-fit using the Hosmer–Lemeshow statistic and other diagnostic tests. The results showed that different factors affect the severity of accidents inside and outside urban areas, including the type of collision, involvement of specific road users, time and location of the accident, and weather conditions. The results of this study can provide a useful tool for prioritizing road safety interventions and improving road safety in Greece and worldwide.

The authors in [40] investigated the impact of traffic and weather characteristics on road safety and how these factors contribute to the occurrence of road accidents. This study provides a comprehensive overview of the existing literature on the topic and identifies gaps in the knowledge that need to be addressed in future studies [40].

Bergel et al. [41] investigated the impact of weather conditions on road accidents with the aim of identifying the relationship between weather factors and road safety. This study applies statistical techniques, such as regression analysis, to analyse the data and determine the effects of different weather conditions, such as precipitation, wind, temperature, and visibility, on the risk of road accidents. The results of this study provide insights into how weather conditions can affect the smooth operation of traffic in road networks [41].

In the literature, the following types of data are collected for the analysis of traffic accidents:

- Accident location: Accident data should include the specific location of each accident, such as the street or intersection where it occurred.
- Accident frequency: Accident data should include the number of accidents that have occurred at each location, as well as the number of accidents that have resulted in fatalities or serious injuries.
- Accident type: Accident data should include information on the type of accident, such as collisions between vehicles, pedestrian accidents, or bicycle accidents.
- Time of day: Accident data should include information on the time of day when the accidents occurred, as this can help to identify patterns and may indicate the presence of factors such as poor lighting or increased traffic volume.
- Weather conditions: Accident data should include information on the weather conditions at the time of each accident, as inclement weather can impact visibility and increase the likelihood of accidents.
- Driver behaviour: Accident data should include information on the behaviour of drivers involved in accidents, such as speeding, distracted driving, and reckless or aggressive behaviour.
- Road design and infrastructure: Accident data should include information on the design and condition of the road where each accident occurred, such as the presence of pedestrian crossings, bike lanes, and road signs.

Other sources of information can be used to identify black spots, including traffic flow, speed, and road-user surveys.

### 3.1. Sources and Resources

Many government agencies, such as transportation departments or road safety agencies, collect and maintain data on accidents and road safety. In the present study many sources were accessed and a plethora of resources were compiled to make the Black Spot Dataset of North Greece (BSNG). For instance, the police, construction agencies, and experts from academia were interviewed to frame what makes a black spot in Greece and what they suspect as the primary causes leading to accidents based on their experience. In addition to the interviews, data were collected through open data portals which are publicly available to users or by requesting relevant agencies for clarifications and any additional information. Note that in Greece, a black spot is considered a stretch of road where two or more accidents have occurred[1,2]. Thus, further analysis was conducted to detect the black spots.

Much of the data comprising the BSNG came from the web portal, Hellenic Statistical Authority (ELSTAT)[3]. To acquire complementary information from the ELSTAT, official request forms were filed and anonymised data were obtained. In Greece this agency operates as an independent organization and has full administrative and financial freedom. Its mission is to safeguard and publish the country's statistics. Collecting data regarding traffic accidents is one amongst the many topics that ELSTAT surveys. ELSTAT acquires data directly from the Greek police. When the Greek police personnel perform an autopsy at the scene of a traffic accident it fills out a report, known as Road Traffic Collision Reports (RTC). These documents (not in electronic form) are sent to ELSTAT, where they are coded and inserted into a database. RTCs can be valuable sources of information because they include information on the type of accident that occur in a specific location, such as collisions between vehicles, pedestrian accidents, and bicycle accidents. In addition to information on the type of accident, police reports typically provide data on the time of day and the weather conditions under which the accident occurred. Information on the age, gender, and driving experience of drivers involved in accidents is also provided, which can help identify demographic groups that may be at a higher risk of accidents in specific areas. Hence, data collected in these reports can help to identify areas of the road network where accidents occur frequently and can provide insight into the underlying causes of these accidents.

*3.2. Dataset Profile*

An eight-step procedure was followed during the process of creating the BSNG, including (i) objective definition, (ii) data organization, (iii) data cleaning, (iv) data preprocessing, (v) data anonymisation, (vi) data review , (vii) analytical documentation, and (viii) data partitioning. For this work, data on road traffic accidents resulting in injury or death were collected from the national and rural network for the regions of Macedonia and Thrace (17 Regional Units) between 2014 and 2018. Thus, the objective of the BSNG is stated as: "To create a data collection for the identification of blackspots and the assessment of factors present in these locations regarding the traffic collisions situated in North Greece between 2014 and 2018".

Information drawn from interviews had to be quantified regarding the degrees of injury, the visibility of vertical and horizontal traffic signs, the road surface specification, procedures to measure the road gradient and, most importantly, the identification of collision sites given that in many cases the reference was bidirectional and not concise. To organise the data into a structured format the use of spreadsheets were employed, with rows representing records and columns representing attributes or features. The following features were extracted, grouped and organised for each accident:

- Accident location;
- Incident and road environment details (month, week of year, number of deaths, serious injuries, minor injuries, total number of injuries, number of vehicles involved, road surface type, atmospheric conditions, road surface conditions, road marking, lane marking, road width, road narrowness, turn sequence, road gradient, straightness, right turn, left turn, boundary line marking left and right, accident severity, type of first collision);
- Driver information (gender and age);
- Vehicle information (type, age, and mechanical inspection status).

Duplicate values were identified and removed. A very small percentage of the data records exhibited missing values. Interpolation between said records and their closest neighbour was applied to maintain the integrity of the dataset. Fifty-four cases were found that had too many missing features; they were discarded promptly. Data cleaning was followed by a data prepossessing step which involved scaling numerical values and the encoding non-numerical features. Qualitative features were assigned categorical labels, whilst quantitative features were normalised. All transformations were documented analytically.

Finally, it is important to mention that special care was given in anonymizing the data records, such that any personal information is concealed and tracing the connection between an individual that participated in a collision and a data record in the BSNG is effectively eliminated. To ensure that personally identifiable information (PII) was removed, individual data points were aggregated into groups to hide individual-level information, such as location specifics, and specific values were replaced with broader categories, for example, the exact ages were replaced with age ranges, namely, 1–25, 26–64 and >65.

All thirty-five (35) variables and their respective mean, minimum and maximum values are shown at Table 2. The number of black spots was determined by applying the guidelines indicated officially by the law. From the total of 1811 accidents, only 142 were situated at black spots (or 1/5th of the 735 unique accident locations), as shown in Table 3.

**Table 2.** Dataset profile after transformations of the extracted data from ELSTAT.

| Variable Name | Data Type | Description | Mode | Mean |
|---|---|---|---|---|
| Year | Categorical | The year of the accident | 2016 | - |
| Month | Categorical | The month of the accident | July | - |
| Weekday | Categorical | The day of the week | Monday | - |
| Week of Year | Numerical | The week number within a year | - | 27.73 |
| Time | Numerical | The time of the accident (in hours) | - | 13.353 |
| Daylight | Binary | Indicates if the accident occurred during daylight or not | Yes | - |
| Deceased | Numerical | The number of deceased individuals in the accident | - | 0.202 |
| Serious injuries | Numerical | The number of individuals with serious injuries | - | 0.168 |
| Minor injuries | Numerical | The number of individuals with minor injuries | - | 1.248 |
| Totally injured | Numerical | The total number of injured individuals | - | 1.248 |
| Vehicles involved | Numerical | The number of vehicles involved in the accident | - | 1.457 |
| Traffic class | Ordinal | The traffic class of the road | 0–1000 | - |
| Roadway type | Categorical | The type of roadway | Tarmac | - |
| Atmospheric conditions | Categorical | The atmospheric conditions during the accident | Good weather | - |
| Roadside environment | Categorical | The environment alongside the road | Habited | - |
| Road surface conditions | Categorical | The condition of the road surface | Normal | - |
| Lane divider | Binary | Presence of a lane divider | Not present | - |
| Road width | Numerical | The width of the road (in meters) | - | 8 |
| Road narrowness | Binary | Indicates if the road is narrow or not | Narrowing | - |
| Lane direction sign | Binary | Presence of lane direction signs | Visible | - |
| Sequential turns | Binary | Sequential turns in the road | False | - |
| Road gradient | Categorical | The gradient or slope of the road | Uphill | - |
| Straightness | Categorical | The straightness of the road | Straight | - |
| Right turn | Binary | Indicates if a right turn was involved in the accident | False | - |
| Left turn | Binary | Indicates if a left turn was involved in the accident | False | - |
| Left barrier | Binary | Presence of a barrier on the left side of the road | Non existent | - |
| Right barrier | Binary | Presence of a barrier on the right side of the road | Non existent | - |
| Left edge line | Categorical | Presence of an edge line on the left side of the road | Not present | - |
| Right edge line | Categorical | Presence of an edge line on the right side of the road | Not present | - |
| Accident severity | Categorical | The severity level of the accident | Wounded | - |
| Vehicle age | Ordinal | Age category of the involved vehicle | 13–15 | - |
| Vehicle type | Categorical | The type of vehicle involved in the accident | Private car | - |
| Mechanical inspection | Categorical | Indicates if a mechanical inspection was conducted | Passed | - |
| Driver's gender | Categorical | The gender of the driver involved in the accident | Female | - |
| Driver's age | Ordinal | The age category of the driver | 26–64 | - |
| Black Spot | Binary | Indicates if the accident occurred at a blackspot | Non-blackspot | - |

See Appendix A, for more information about the classes.

**Table 3.** Traffic accidents and black spot locations in Macedonia and Thrace: Split by regional units.

| Regional Unit | Traffic Accidents | Black Spots |
|---|---|---|
| Thessaloniki | 369 | 32 |
| Chalkidiki | 272 | 21 |
| Xanthi | 103 | 13 |
| Serres | 150 | 12 |
| Evros | 120 | 11 |
| Rhodopi | 108 | 11 |
| Kilkis | 127 | 9 |
| Pella | 99 | 8 |
| Thasos | 54 | 5 |
| Drama | 33 | 4 |
| Imathia | 69 | 4 |
| Kavala | 124 | 4 |
| Kozani | 65 | 4 |
| Pieria | 46 | 1 |
| Kastoria | 36 | 1 |
| Grevena | 24 | 1 |
| Florina | 12 | 1 |
| **Summary** | **1811** | **142** |

## 4. Benchmarking

### 4.1. Statistical Modelling

4.1.1. Poisson Distribution

The Poisson's distribution is a probability distribution that is often used in statistical analysis to model the occurrence of events over a specified time period or in a specific area. It is named after French mathematician Siméon Denis Poisson, who first introduced the distribution in the early 19th century. It is particularly useful for modelling rare events that occur randomly and independently of each other, such as accidents, failures, or defects in a manufacturing process. The distribution is characterized by a single parameter, denoted as lambda ($\lambda$), which represents the expected number of events that occur over a given time or area.

The probability of observing *k* events in a Poisson distribution is given by the formula:

$$P(k) = \frac{e^{-\lambda} * \lambda^k}{k!} \tag{1}$$

where *e* is the mathematical constant approximately equal to 2.718, and *k*! is the factorial of *k*. It is important to note that while the Poisson distribution is unbounded, it does become increasingly unlikely to observe values much larger than the mean. This is because the Poisson distribution has a property called "overdispersion", which means that the variance of the distribution is larger than the mean. As a result, extremely large values are relatively rare, even if the mean is quite large.

Poisson regression is a type of generalized linear model that is used to model count data, where the response variable is a count of events occurring in a fixed period of time or within a specified region. The Poisson distribution is used to model the probability of observing a given number of events in a fixed time interval, assuming that the events occur independently at a constant rate. The Poisson regression model extends this idea by allowing us to model the relationship between the mean and the explanatory variables.

Let $Y_i$ be the observed count for the *i*th observation, and let $x_{ij}$ be the value of the *j*th predictor variable for the *i*th observation. The Poisson regression model assumes that:

$$Y_i \sim \text{Poisson}(\lambda_i) \tag{2}$$

where $\lambda_i$ is the expected count for the *i*th observation, and is modelled as:

$$\log(\lambda_i) = \beta_0 + \beta_1 x_{i1} + \cdots + \beta_p x_{ip} \tag{3}$$

where $\beta_0, \beta_1, \ldots, \beta_p$ are the regression coefficients to be estimated, and *p* is the number of predictor variables. The use of the logarithmic function on the right-hand side ensures that $\lambda_i$ is non-negative. Thus, Poisson modelling can lose its accuracy over time due to changes in the underlying data generating process or the road network itself. For example, changes in traffic volume, road design, and infrastructure can impact the distribution of accidents and make Poisson models trained on historical data less accurate over time. In addition, improvements in vehicle safety technology and changes in driver behaviour can also impact the accuracy of Poisson models.

4.1.2. Naive Bayes

Naive Bayes is a simple and effective probabilistic algorithm used in machine learning for classification tasks. It is based on the Bayes' theorem and the assumption of conditional independence between the features given the class. In simple terms, it assumes that the occurrence of one feature is not related to the occurrence of any other feature. The algorithm works by first learning the probabilities of the different classes and the probabilities of the different features given each class, using a training dataset. Then, for a new input data point, the algorithm calculates the probability of each class given the observed features, using the learned probabilities.

Mathematically, the naive Bayes algorithm can be formulated as follows: Given a set of input features $x = (x_1, x_2, ..., x_n)$, the algorithm calculates the probability of each class $c$, given the observed features, using Bayes' theorem:

$$P(c|x) = P(x|c) \cdot P(c)/P(x) \tag{4}$$

where $P(c|x)$ is the probability of class $c$ given the observed features $x$. $P(x|c)$ is the conditional probability of observing the features $x$ given the class $c$. $P(c)$ is the prior probability of class $c$ and $P(x)$ is the evidence probability of the features $x$.

The Naive Bayes algorithm makes the assumption of conditional independence between the features given the class, which simplifies the calculation of $P(x|c)$ to the product of the probabilities of each feature given the class:

$$P(x|c) = P(x_1|c) * P(x_2|c) * ... * P(x_n|c) \tag{5}$$

This allows the algorithm to estimate the probabilities of the features given each class independently, which reduces the number of parameters to be learned and makes the algorithm computationally efficient.

### 4.1.3. Gaussian Process

A Gaussian process (GP) is a powerful, non-parametric Bayesian method for regression and classification tasks in machine learning. It is based on the concept of a collection of random variables, where any finite subset of these variables has a joint Gaussian distribution. In the context of machine learning, Gaussian processes are used to model the relationship between input features and output values as a probability distribution over functions.

Specifically, a GP defines a prior distribution over functions, which captures the belief about the function space before observing any data. This prior distribution is characterized by a mean function and a covariance function (also known as a kernel function). The mean function usually defaults to zero, while the kernel function encodes the properties of the functions, such as smoothness and correlation between input points. The kernel function, $K(x, x')$, quantifies the similarity between two input points $x$ and $x'$ in the feature space. It plays a crucial role in determining the properties of the GP. Common kernel functions include the radial basis function (RBF), Matern, and exponential kernels. The choice of the kernel function and its parameters significantly influences the performance of the GP. After observing the training data, the GP computes the posterior distribution over functions, which combines the prior distribution with the observed data. The posterior distribution encodes the updated belief about the function space, given the observed data points. Finally, to make predictions for a new input point $x*$, the GP computes the conditional distribution of the output value $y*$ given the observed data and the input point. The prediction includes both the mean and the variance, providing a measure of uncertainty associated with the prediction. This uncertainty estimate is a key advantage of GPs over other machine learning methods.

Regarding the tuning the GP, the kernel function typically has hyperparameters that control its properties, such as the length scale and output variance. These hyperparameters can be learned from the data by optimizing the marginal likelihood, which is the likelihood of the observed data given the GP model.

In summary, GPs are a flexible and powerful Bayesian method for modelling the relationship between input features and output values in regression and classification tasks. They provide a probabilistic framework that can capture the uncertainty associated with predictions and can be adapted to various types of data by choosing appropriate kernel functions and optimizing their hyperparameters.

### 4.2. Nearest Neighbours

The k-nearest neighbours (k-NN) algorithm is a non-parametric, lazy learning method used for both classification and regression tasks in machine learning. It is based on the

idea that similar data points in a feature space should have similar labels or outputs (see Algorithm 1).

---

**Algorithm 1** k-Nearest Neighbors Algorithm.

---

　　**procedure** KNN($x_{tr}, q, k, D$)
　　　　Initialize an empty list $L$
　　　　**for** $i = 1$ to $N$ **do**
　　　　　　$d \leftarrow$ D($x_{tr}[i], q$)
　　　　　　Append ($d, x_{tr}[i].label$) to $L$
　　　　**end for**
　　　　Sort $L$ by the first element (the distance values)
　　　　Select the first $k$ elements of $L$, denoted as $nn$
　　　　**if** classification task **then**
　　　　　　Return the majority class label among the $nn$
　　　　**else if** regression task **then**
　　　　　　Return the average output of the $nn$
　　　　**end if**
　　**end procedure**

---

The k-NN algorithm operates in a feature space, where each initial data point is represented as a point in a multidimensional space. Each dimension of the space corresponds to a feature of the data. To determine the similarity between data points, a distance metric is used. Common distance metrics include Euclidean distance, Manhattan distance, and cosine similarity. The choice of distance metric depends on the nature of the data and the problem. The $k$ in k-NN refers to the number of nearest neighbours considered in the algorithm. A higher value of $k$ results in a more complex decision boundary and increased smoothness, while a lower value of $k$ results in a more flexible and potentially overfitting decision boundary.

### 4.3. Support Vector Machines

Linear SVM

Support vector machine (SVM) is a supervised learning algorithm used for classification and regression tasks. The linear SVM specifically focuses on linearly separable problems, where the classes can be separated by a linear decision boundary or hyperplane. The primary goal of the linear SVM is to find the optimal hyperplane that maximizes the margin between the two classes.

The objective is defined as: Given a set of labelled training data, a linear SVM aims to find the optimal separating hyperplane that maximizes the margin between the classes while minimizing the classification error. The linear SVM assumes that the data is linearly separable, meaning that a straight line (in two dimensions), a plane (in three dimensions), or a hyperplane (in higher dimensions) can separate the classes without any errors. The margin is defined as the distance between the hyperplane and the closest data points from each class. These closest data points are called support vectors. The linear SVM seeks the hyperplane that maximizes this margin, as it provides the most robust separation between the classes. Furthermore, the linear SVM can be formulated as a convex optimization problem, where the objective is to minimize the norm of the weight vector $||w||$ while satisfying the constraints that the data points are correctly classified with a margin of at least one.

$$\min_{w,b} \frac{1}{2}||w||^2, \quad y_i(w \cdot x_i + b) \geq 1, ; i = 1, \ldots, N \tag{6}$$

$w$ represents the weight vector that defines the hyperplane, $b$ is the bias term, $x_i$ are the feature vectors, $y_i$ are the class labels ($+1$ or $-1$), and $N$ is the number of training data points.

To solve the constrained optimization problem, Lagrange multipliers are used to convert it into the dual problem, which is a maximization problem.In cases where the data are not perfectly linearly separable, a soft margin SVM can be used. This allows some misclassifications in exchange for a larger margin, which can be controlled by a regularization parameter $C$. This helps balance the trade-off between maximizing the margin and minimizing the classification error.

### 4.4. Tree-Based Methods

#### 4.4.1. Decision Tree

A decision tree is a supervised machine learning algorithm used for both classification and regression tasks. It models the relationship between input features and output values using a tree-like structure, where each internal node represents a decision rule based on a feature value, and each leaf node represents the predicted output value. The algorithm recursively splits the input feature space into distinct regions, allowing it to capture non-linear relationships and interactions between features.

In detail, a decision tree consists of internal nodes (decision nodes), branches, and leaf nodes (terminal nodes). Each internal node represents a decision rule based on a feature value, each branch corresponds to the outcome of the decision rule, and each leaf node represents the predicted output value (class label for classification or a numerical value for regression).

The algorithm recursively splits the input feature space into distinct regions, with each split being determined by a feature value. This process continues until a stopping criterion is met, such as reaching a maximum tree depth or a minimum number of samples per leaf node. To decide the best feature and value to split the data at each node, the algorithm uses a split criterion. For classification tasks, common split criteria include Gini impurity, information gain, and the chi-squared statistic.

Decision trees have a tendency to overfit the training data, resulting in poor generalization to new, unseen samples. To mitigate overfitting, techniques such as pruning are used to reduce the complexity of the tree. Pruning can be performed during the tree construction (pre-pruning) by setting limits on tree depth, minimum samples per leaf, or minimum impurity decrease. Alternatively, pruning can be performed after the tree construction (post-pruning) by removing or collapsing branches that do not significantly improve the model's performance on a validation set.

One of the key advantages of decision trees is their interpretability. The tree structure and decision rules can be easily visualized and understood by humans, making them a popular choice in situations where model explainability is crucial.

#### 4.4.2. Random Forest

Random forest is an ensemble learning method used for both classification and regression tasks in machine learning. It builds multiple decision trees and combines their predictions to improve accuracy and reduce overfitting. The aggregation of the predictions of multiple trees allows random forest to capture complex interactions between features and provide a more robust and accurate model compared to a single decision tree.

Regarding how it works, a random forest constructs multiple decision trees, each trained on a random subset of the data. The final prediction is made by averaging the predictions (for regression) or taking a majority vote (for classification) of the individual trees. For each decision tree, a random subset of the training data can be selected with a replacement (also known as bootstrapping). This process results in different trees being trained on slightly different data, aiming at increasing the diversity of the trees and reducing the overfitting phenomena.

Another technique to battle overifitting is feature bagging. At each split in the decision tree, a random subset of features is considered for splitting, instead of using all available features. This further increases the diversity of the trees and helps in capturing complex interactions between features.

Finally, to make a prediction for a new input, the input is passed through each decision tree in the ensemble. The class with the majority vote among the trees is chosen as the final prediction. Since each tree can be trained on a bootstrapped subset of the data, some samples are not used in the training process for a given tree. These unused samples, known as out-of-bag samples, can be used as a validation set to estimate the model's performance without the need for a separate validation set.

Random forest can provide an estimate of variable importance by calculating the average decrease in the impurity (Gini impurity for classification, mean squared error for regression) when a given feature is used for splitting across all trees in the forest.

### 4.4.3. Extra Randomised Trees

Extra randomized trees, also known as extremely randomized trees or ExtraTrees, is an ensemble learning method used for classification and regression tasks in machine learning. Similar to random forest, it builds multiple decision trees and combines their predictions to improve accuracy and reduce overfitting. The key difference between ExtraTrees and random forest lies in the way the individual trees are constructed.

The ExtraTrees algorithm constructs multiple decision trees, with the final prediction made by taking a majority vote (for classification) of the individual trees. It can be trained on either the full dataset or a bootstrapped subset of the data (with replacement), similar to random forest. The choice depends on the specific implementation and the desired trade-off between bias and variance. The main difference between ExtraTrees and other tree-based approaches is in the way the individual trees are constructed. In ExtraTrees, at each split in the decision tree, a random subset of features is considered for splitting, and the best split threshold is chosen randomly among the possible thresholds for each feature. This additional layer of randomness results in trees that are less prone to overfitting and as diverse as they can be.

To make a prediction for a new input, the input is passed through each decision tree in the ensemble. For classification tasks, the class with the majority vote among the trees is chosen as the final prediction. This approach can provide an estimate of variable importance by calculating the average decrease in impurity (Gini impurity for classification) when a given feature is used for splitting across all trees in the forest.

### 4.4.4. AdaBoost

AdaBoost, short for adaptive boosting, is a machine learning algorithm that is used for classification tasks. It was introduced by Yoav Freund and Robert Schapire in 1997 as a method for boosting the performance of weak classifiers by combining them into a strong, ensemble classifier.

The basic idea behind AdaBoost is to iteratively train a series of weak classifiers, typically decision trees, on a given dataset, and then combine their predictions to create a final, strong classifier. The weak classifiers are trained on different subsets of the data, and at each iteration, the weights of the misclassified samples are increased. This ensures that subsequent classifiers focus more on the misclassified samples, thus improving the overall performance of the ensemble.

AdaBoost has proven to be effective in various applications and is considered one of the earliest and most popular ensemble learning methods. Its main advantages include its simplicity, adaptability, and strong performance with various types of classifiers and datasets. However, it can be sensitive to noise and outliers in the data, as these can lead to an increased focus on misclassified samples during the training process.

### 4.5. Multilayered Perceptron

A multilayer perceptron (MLP) is a type of artificial neural network used for classification and regression tasks in machine learning. It consists of multiple layers of nodes (neurons) organized in a feed-forward structure, with each layer fully connected to the next one. MLPs are capable of learning complex, non-linear relationships between input

features and output values by adjusting the weights of the connections through a process called backpropagation.

An MLP typically consists of an input layer, one or more hidden layers, and an output layer. Each layer consists of multiple nodes, where the input layer corresponds to the input features, the hidden layers perform non-linear transformations, and the output layer produces the final predictions.

The nodes in the hidden layers use an activation function to introduce non-linearity into the model. Common activation functions include the sigmoid, hyperbolic tangent (tanh), and rectified linear unit (ReLU). Given an input vector $x$, the MLP computes the output through a process called forward propagation. For each layer $l$, the pre-activation value ($a$) and the activation value ($z$) are computed as follows:

$$a^{(l)} = W^{(l)}z^{(l-1)} + b^{(l)} \tag{7}$$

$$z^{(l)} = f(a^{(l)}) \tag{8}$$

where $W^{(l)}$ is the weight matrix for layer $l$, $b^{(l)}$ is the bias vector for layer $l$, $f$ is the activation function, and $z^{(0)} = x$.

The output layer uses an output function to produce the final predictions. For classification tasks, the softmax function is commonly used to compute class probabilities, while for regression tasks, a linear output function is typically used. To train the MLP, a loss function is used to measure the discrepancy between the predicted values and the true output values. Common loss functions include the cross-entropy loss for classification and the mean squared error for regression. The weights and biases in the MLP are adjusted using an optimization algorithm, usually gradient descent or a variant thereof, in combination with the backpropagation algorithm. Backpropagation computes the gradients of the loss function with respect to the weights and biases by applying the chain rule of calculus. The gradients are then used to update the weights and biases to minimize the loss function.

## 5. Proposed Deep Learning Method

Various statistical methods, such as the Poisson and negative binomial models, both in univariate and multivariate frameworks, have been applied with success in analysing crash counts. These approaches aim to address the data and methodological challenges associated with estimating and predicting traffic crashes and deepen our understanding of the relationship between influencing factors and crash outcomes [42]. However, current research in traffic safety shows that these statistical models need to be revised to deal with complex, highly non-linear data, indicating that the relationship between influencing factors and crash outcomes is more intricate than what can be captured by a single statistical method. For this reason, the proposed approach uses a deep learning framework to tackle the otherwise inexplicable non-linear relationships among the variables.

However, it is essential to note that the Poisson distribution assumes that events occur independently of each other and that the probability of an event occurring is proportional to the length of the time interval. If these assumptions do not hold, the Poisson distribution may not be an appropriate model. Especially when modelling rare occurrences, it is also important to consider the amount and quality of the available data. If there are only a few occurrences of an event, it may not be easy to accurately estimate the distribution parameters, and the resulting model may not be reliable. In such cases, alternative methods such as Bayesian inference or machine learning may supplement or replace traditional statistical models.

Police reports for black spot analysis can contain both qualitative and quantitative data. Qualitative data refers to descriptive information, such as descriptions of road conditions, driver behaviour, and weather conditions. Quantitative data refers to numerical data, such as the number of accidents, number of vehicles involved, and speed limits. Both types of data can be important in identifying the causes and contributing factors of road

accidents and can be used in the analysis of black spots. Yet, machine learning algorithms do not handle mixed data well and typically use a combination of techniques to deal with them. There are several techniques that can be used to handle each type of data in machine learning, such as one-hot encoding for encoding categorical data to a numerical representation, ordinal encoding to map qualitative data into numerical values based on the order of the categories, and label encoding to map categories into a unique numerical value. These techniques ensure that the data are in a format that can be used effectively in the algorithms, and can help to improve the accuracy and performance of the models.

### 5.1. High Level Overview

The proposed approach is divided into the following four steps. At the beginning, each variable is transformed into labelled and one-hot encodings, to ensure the applicability of the machine learning methods. Then a self-supervised deep learning architecture is used to reduce the dimensions of the input features into a compact latent vector. An augmentation method, such as MixUp, is used to linearly combine the feature vectors of the samples of the blackspot class to even the number of samples of the two classes of the BSNG dataset. The latent vectors are used as the input into a classifier to approximate a binary class problem, e.g., whether the sample is a black spot or not. A graphical overview of the proposed method is shown in Figure 1.

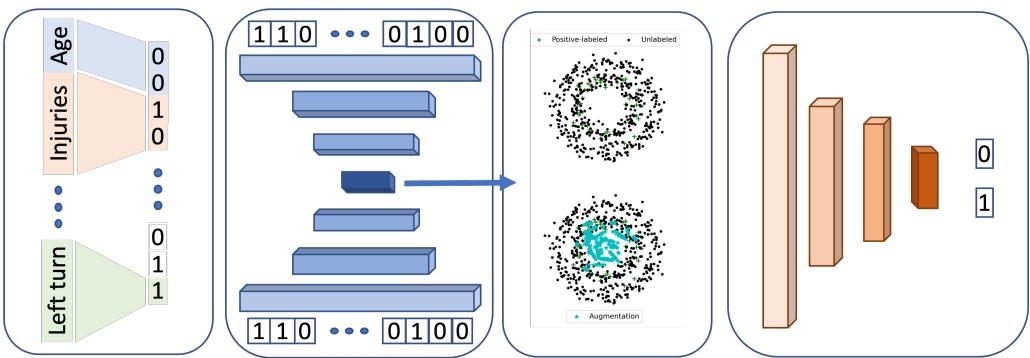

**Figure 1.** A high-level overview of the proposed method for the identification of black spots in the BSNG dataset.

### 5.2. One-Hot Encoding

One-hot encoding is a widely used technique for encoding categorical variables in machine learning and data analysis. Categorical variables are variables that can take on a limited set of values, such as colours, types of animals, or car brands. Since many machine learning algorithms cannot process categorical variables directly, we need to convert them to numerical values before using them in models.

One-hot encoding is a way of representing categorical variables as numerical values in a binary format. This technique creates a binary vector where each column corresponds to a possible category, and only one of the columns is "hot" or "on" at a time, indicating the presence of that category in the input data.

In practical terms, suppose a weather dataset contains a column named "Weather" that has four possible values: "sunny", "cloudy", "rainy", and "snowy". Using one-hot encoding, the "Weather" column would be transformed into four binary columns, with each row containing a single "1" value in the column that corresponds to the weather condition of that row (see Table 4).

One-hot encoding has several advantages over other encoding techniques. For one, it preserves the information in the original categorical variable and does not introduce any artificial ordering or ranking of the categories. Additionally, it allows for easy comparison of different categories since they are represented by separate columns, and it works well with many machine learning algorithms.

**Table 4.** An example of one-hot encoding weather conditions

| Original | | Encoded | | | |
|---|---|---|---|---|---|
| Index | Condition | Sunny | Cloudy | Rainy | Snowy |
| 1 | Sunny | 1 | 0 | 0 | 0 |
| 2 | Cloudy | 0 | 1 | 0 | 0 |
| 3 | Sunny | 1 | 0 | 0 | 0 |
| 4 | Rainy | 0 | 0 | 1 | 0 |
| 5 | Snowy | 0 | 0 | 0 | 1 |

### 5.3. Self-Supervised Learning

An autoencoder is a type of unsupervised artificial neural network used for dimensionality reduction, feature learning, and data compression. It consists of two main components: an encoder that maps the input data to a lower-dimensional latent representation, and a decoder that reconstructs the input data from the latent representation. The autoencoder learns to compress and reconstruct the input data by minimizing the difference between the input and the reconstructed output, often using a loss function such as the mean squared error. Both the encoder and decoder can consist of multiple layers.

Given an input vector $x$, the encoder computes the latent representation $z$ through a series of transformations:

$$z = f_E(x) = f_L(\ldots f_2(f_1(x; W_1, b_1); W_2, b_2) \ldots ; W_L, b_L) \tag{9}$$

where $f_i$ is the activation function for layer $i$, $W_i$ and $b_i$ are the weight matrix and bias vector for layer $i$, and $L$ is the number of layers in the encoder.

Given the latent representation $z$, the decoder computes the reconstructed input data $x'$ through a series of transformations:

$$x' = f_D(z) = f_{L+M}(\ldots f_{L+2}(f_{L+1}(z; W_{L+1}, b_{L+1});$$
$$W_{L+2}, b_{L+2}) \ldots ; W_{L+M}, b_{L+M}) \tag{10}$$

where $f_i$ is the activation function for layer $i$, $W_i$ and $b_i$ are the weight matrix and bias vector for layer $i$, respectively, and $M$ is the number of layers in the decoder.

The autoencoder learns to compress and reconstruct the input data by minimizing the difference between the input $x$ and the reconstructed output $x'$. A common loss function for this purpose is the mean squared error (MSE):

$$\mathcal{L}(x, x') = \frac{1}{N} \sum_{i=1}^{N} ||x_i - x_i'||^2 \tag{11}$$

where $N$ is the number of input samples.

The autoencoder is trained using an optimization algorithm, usually gradient descent or a variant thereof, in combination with backpropagation to compute the gradients of the loss function with respect to the weights and biases. The weights and biases are then updated to minimize the loss function.

### 5.4. Augmentation via MixUp

MixUp is a data augmentation technique proposed for training deep learning models, particularly in the context of image classification tasks. MixUp generates new training samples by taking convex combinations of pairs of input samples and their corresponding labels. This technique encourages the model to learn more robust features and improves its generalization performance.

Given a dataset with input samples $X$ and labels $Y$, two samples $x_1$ and $x_2$ are randomly selected with corresponding labels $y_1$ and $y_2$. A mixing parameter $\lambda$ is sampled as

well from a predefined distribution. Typically the beta distribution with parameters $\alpha > 0$ and $\beta > 0$ (e.g., $\alpha = \beta = 0.4$).

A new mixed sample $x'$ and its corresponding label $y'$ are created using the convex combination of $x_1$, $x_2$, $y_1$, and $y_2$ with the mixing parameter $\lambda$:

$$x' = \lambda x_1 + (1 - \lambda)x_2 \tag{12}$$

$$y' = \lambda y_1 + (1 - \lambda)y_2 \tag{13}$$

The new mixed sample $x'$ and its corresponding label $y'$ are appended to the training dataset.

The MixUp technique encourages the model to behave linearly between the training samples, leading to smoother decision boundaries and improved generalization performance. It can be used in combination with other data augmentation techniques, such as random cropping, flipping, or rotations, to further enhance the model's robustness.

## 6. Experiments and Discussion

### 6.1. Configuration

In this work, the BSNG dataset is introduced, a comprehensive collection of collision data that occurred on the national road network of North Greece between 2014 and 2018. This dataset comprises 1811 samples, including 142 black spots, as classified by the ELSTAT. To analyse these black spots, ten classic machine learning methods are employed, as well as a novel approach is proposed. The machine learning methods utilized include Poisson regression, naive Bayes, GPs, k-NN, linear and non-linear SVM, decision tree, ensemble methods such as random forest, extra randomised trees, adaptive boosting, and a neural network. All models and experiments were conducted on the same platform, using Python and Keras as the deep learning framework. This paper aims to provide valuable insights into the analysis of black spots using machine learning techniques, potentially contributing to reducing traffic accidents on the national road network in North Greece.

All methods applied to the BSNG dataset required hyperparameter optimization. In all experiments, hyperparameters were optimized using 5-fold cross-validation, with the aim of maximizing the F1 score. For k-NN, the value of $k$ was set to 4, which is twice the threshold applied by the Greek authorities. Poisson regression was tuned using a regularization strength of $\alpha = 0.9$ and the low memory Broyden–Fletcher–Goldfarb–Shanno (LBFGS) solver with a tolerance of $10^{-5}$. GP modelling required an appropriate kernel to fit the data, and a radial basis kernel (RBF) with a small length scale of $I = 0.125$ was used. The non-linear case of SVM also used an RBF kernel, with $\gamma = \frac{1}{2 \cdot I^2}$. The dimensionality of the data were reduced to five components with PCA, in order for the SVMs to work properly. For tree-based methods, feature bagging was used without pruning. Ensemble methods used 30 estimators. The multilayered perceptron used five fully connected layers with $(512, 7, 64, 32, 4)$ nodes, ReLU activation, a learning rate of $10^{-4}$, the Adam solver, and 100 training epochs. The number of nodes in these layers were discovered by a grid searching approach.

In the following experiments, the original BSNG dataset underwent transformations in an effort to improve the performance of the methods. First, a one-hot encoding was applied to each variable, resulting in a dataset with 687 dimensions, up from the original 35. In the last experiment, the MixUp technique was used to augment the dataset. This involved sampling 6000 random pairs with replacement and creating 11 additional samples for each selected pair of initial samples, using a beta distribution with $\beta(0.2, 0.2)$. This resulted in a total of 67,448 training samples.

The proposed method is unique compared to others because it takes a one-hot encoded BSNG as the input and employs an autoencoder to generate latent representations. Data augmentation is performed using MixUp. The autoencoder's architecture has a bottleneck layout with the encoder consisting of three layers with ReLU activations and nodes of

size $(256, 64, 32)$. The decoder has the reverse layer order. The optimizer used is Adam with a learning rate of $10^{-4}$. MixUp is applied by sampling 6000 random pairs with replacement and creating 11 additional samples for each selected pair of initial samples using a beta distribution with $\beta(0.2, 0.2)$. This resulted in a total of 67,448 training samples. The classifier is an MLP with three fully connected layers with $(32, 24, 6)$ nodes, ReLU activation, a learning rate of $10^{-4}$, the Adam solver, and 100 training epochs. The number of nodes in these layers were discovered by a grid searching approach as before.

*6.2. Results*

The BSNG dataset poses a significant challenge to classification algorithms, as indicated by the findings presented in Table 5. Although high accuracy rates are reported, they are misleading and should not be relied upon. The imbalanced nature of the dataset makes it inappropriate to judge performance based on the fraction of correct predictions. For example, a classifier that assigns all samples to the "non-black spot" class would achieve an accuracy of 87.5%, but would be useless in practice.

**Table 5.** Comparison of performances between the proposed method and textbook methods. (Datasets: A = original, B = one-hot encoded, C = encoded & augmented).

| Dataset | Method | Acc (std) | | Prec (std) | | Rec (std) | | F1 (std) | | AUC (std) | |
|---|---|---|---|---|---|---|---|---|---|---|---|
| | Poisson | 74.93 | (3.20) | 19.14 | (1.67) | 14.51 | (2.08) | 16.51 | (2.54) | 50.94 | (4.12) |
| | Naive Bayes | 56.74 | (4.06) | 18.95 | (2.98) | 46.77 | (2.70) | 26.97 | (4.03) | 52.78 | (3.85) |
| | Gaussian Process | 69.14 | (2.81) | 15.27 | (3.02) | 17.74 | (3.22) | 16.41 | (3.14) | 48.73 | (2.67) |
| | kNN | 68.31 | (2.85) | 14.66 | (2.33) | 27.74 | (3.18) | 16.05 | (2.63) | 48.23 | (2.98) |
| | Linear SVM | 68.04 | (3.12) | 14.28 | (1.98) | 0.16 | (0.89) | 0.28 | (1.06) | 49.80 | (2.77) |
| A | Decision Tree | 76.03 | (2.72) | 30.15 | (2.09) | 30.64 | (2.78) | 30.41 | (2.63) | 58.01 | (3.12) |
| | Random Forest | 80.71 | (1.94) | 40.01 | (2.48) | 25.81 | (2.09) | 31.37 | (2.63) | 58.91 | (1.98) |
| | Xtra Trees | 77.96 | (2.47) | 33.33 | (2.73) | 29.03 | (2.28) | 31.03 | (2.67) | 58.53 | (2.42) |
| | AdaBoost | 53.16 | (3.34) | 17.07 | (2.89) | 45.16 | (2.18) | 25.92 | (3.14) | 51.65 | (4.21) |
| | MLP | 79.61 | (2.47) | 25.00 | (1.83) | 10.67 | (0.75) | 13.96 | (1.01) | 51.84 | (2.11) |
| | Poisson | 70.24 | (0.83) | 14.62 | (1.27) | 14.51 | (0.92) | 14.28 | (1.04) | 48.12 | (1.11) |
| | Naive Bayes | 48.23 | (0.62) | 21.19 | (0.87) | 79.03 | (1.23) | 34.26 | (0.94) | 60.44 | (1.08) |
| | Gaussian Process | 79.33 | (1.01) | 21.73 | (0.84) | 0.181 | (0.06) | 19.54 | (1.02) | 51.04 | (0.92) |
| | kNN | 71.62 | (1.14) | 16.39 | (0.79) | 16.12 | (0.98) | 16.26 | (1.01) | 49.59 | (0.89) |
| | Linear SVM | 82.92 | (0.98) | 50.01 | (1.33) | 30.64 | (0.96) | 28.02 | (0.91) | 61.16 | (1.22) |
| B | Decision Tree | 73.55 | (1.02) | 20.68 | (0.91) | 19.35 | (0.83) | 19.99 | (0.95) | 52.03 | (0.99) |
| | Random Forest | 80.16 | (0.94) | 38.63 | (1.12) | 27.41 | (0.79) | 32.07 | (0.87) | 59.22 | (1.09) |
| | Xtra Trees | 81.26 | (1.05) | 43.24 | (0.98) | 25.81 | (0.86) | 32.32 | (1.01) | 59.41 | (1.11) |
| | AdaBoost | 54.26 | (0.92) | 17.81 | (0.77) | 43.28 | (0.99) | 24.54 | (0.88) | 50.01 | (0.91) |
| | MLP | 28.65 | (0.78) | 18.32 | (0.89) | 91.93 | (1.34) | 30.56 | (0.93) | 53.77 | (1.01) |
| | Poisson | 36.63 | (2.50) | 13.79 | (1.08) | 51.61 | (3.20) | 21.76 | (2.11) | 44.49 | (2.58) |
| | Naive Bayes | 50.13 | (2.91) | 22.58 | (1.15) | 79.03 | (3.36) | 35.12 | (2.44) | 61.69 | (4.72) |
| | Gaussian Process | 66.94 | (2.30) | 20.40 | (1.59) | 32.25 | (2.98) | 25.00 | (2.29) | 53.27 | (3.13) |
| | kNN | 63.25 | (2.02) | 14.85 | (1.56) | 24.19 | (2.26) | 18.42 | (1.90) | 47.81 | (2.60) |
| | Linear SVM | 68.61 | (2.45) | 26.36 | (2.12) | 46.77 | (2.84) | 33.72 | (2.50) | 59.93 | (3.35) |
| C | RBF SVM | 81.81 | (3.10) | 43.75 | (2.63) | 22.25 | (1.89) | 29.78 | (2.29) | 58.31 | (3.70) |
| | Decision Tree | 69.42 | (2.21) | 21.83 | (1.82) | 30.64 | (2.92) | 25.52 | (2.08) | 54.02 | (2.73) |
| | Random Forest | 79.33 | (2.74) | 37.73 | (2.31) | 32.25 | (2.25) | 34.78 | (2.42) | 60.64 | (3.14) |
| | Xtra Trees | 82.36 | (2.83) | 45.45 | (2.57) | 16.12 | (1.71) | 24.44 | (2.03) | 56.07 | (2.79) |
| | AdaBoost | 69.69 | (2.41) | 26.11 | (2.06) | 41.93 | (2.68) | 32.09 | (2.93) | 58.67 | (3.00) |
| | MLP | 78.23 | (2.62) | 36.92 | (2.63) | 38.79 | (2.54) | 37.77 | (2.49) | 62.54 | (3.21) |
| | **Proposed** | **84.02** | (0.49) | **52.85** | (1.51) | **59.67** | (0.99) | **56.06** | (1.31) | **74.35** | (1.92) |

To provide a more complete picture of performance, we present four additional metrics: precision, recall, $F_1$-score, and area under the curve (AUC). Using the dataset as-is, results in poor classification performance for all methods, due to the imbalance between the two classes and the difficulty of identifying the features that distinguish black spots from non-black spots. Encoding the feature space, which contains both continuous and categorical variables, improves performance slightly, but the overall performance remains subpar due to the limited number of samples in the dataset.

To address this issue, we augment the dataset with virtual samples created using the MixUp technique. This approach improves the classification performance of all methods, particularly when the feature space is small. However, statistical modelling appears inadequate for handling larger feature spaces and low sample counts. In contrast, tree-based ensemble methods outperform other approaches in terms of $F_1$-score and AUC.

The proposed method, which utilizes a shallow MLP and the low dimensionality of the latent features obtained through MixUp, outperforms all other methods on all metrics. Although the black spot detection scores are low by modern standards, they are comparable to the results reported in [39], which achieved an accuracy of 68.95%, a precision of 52.1%, a recall of 77%, an AUC of 64.1%, and an $F_1$-score of 62.14%. Overall, the findings of this work underscore the importance of carefully considering dataset characteristics and choosing appropriate metrics when evaluating classification algorithms.

### 6.3. Discussion

In the field of statistical modelling, many approaches rely on strict assumptions, such as pre-determined error distributions, and often struggle to handle issues such as multicollinearity and noisy or missing data [42]. To address these challenges, ensemble methods such random forest and AdaBoost have gained popularity. Random forest is an ensemble of decision trees, each trained on a random subset of the data and features, and the final prediction is made by aggregating the individual tree predictions. In contrast, AdaBoost iteratively trains weak classifiers, typically shallow decision trees, by adjusting weights of misclassified samples. The final prediction is made by taking a weighted majority vote of the weak classifiers.

In the context of the BSNG dataset, random forest and XtraTrees performed better than AdaBoost in identifying accidents that occurred in black spots. Increasing the sample population through techniques such as MixUp also helped improve the performance of most algorithms. The proposed method in this study uses an alternative encoding technique and augmented the dataset with virtual samples created through MixUp. In the experimental design, the performance of the proposed method was compared to other setups that included using a textbook encoding technique, such as one-hot encoding, and augmenting the dataset with MixUp. The results show that performance gains were achieved when increasing the sample population and using the proposed method's encoding technique. The proposed method outperformed all other setups in terms of accuracy, precision, recall, $F_1$-score and AUC. This suggests that the proposed method can be a promising solution for identifying accidents that occurred in black spots, despite the challenges posed by the imbalanced and complex BSNG dataset.

Previous studies have shown mixed effects of traffic flow on accident rates, with some suggesting a non-linear correlation and others indicating a linear relationship [40,43]. The impact of traffic flow and congestion on black spot determination still remains uncertain. Similarly, weather patterns seemed to vary and low visibility consistently influenced road safety, as expected.

It is noteworthy to mention that some findings of this research diverge from the prevalent understanding of the main causes of accidents involving young or novice drivers, as highlighted in an earlier study [44]. Contrary to the reference which examined the period from 1995 to 2001 in Greece, the results of the current study indicate that non-compliant behaviour and lack of driving experience may not be the primary contributing factors to accidents in the region under investigation.

According to the data collected, it was found that a mere 3.51% of the vehicles involved in black spot accidents during the specified period did not possess a valid technical control certificate. This finding suggests that the issue of non-compliance with vehicle technical regulations may not have a substantial impact on accident occurrence. Furthermore, the analysis revealed that the age group most frequently involved in accidents at black spots was not composed of young or novice drivers, but rather individuals driving cars

that were more than 10 years old. In fact, these older vehicles accounted for approximately 70% of all accidents that transpired in black spots within the study area.

These unexpected findings challenge the previously established notions regarding the main causes of accidents in the region, as well as the assumptions underlying the recent changes in traffic enforcement policies. The aforementioned research study [44] conducted in Greece highlighted the necessity for stricter police enforcement to address non-compliant behaviour among drivers, while our findings do not necessarily contradict this recommendation, they shed light on the fact that other factors may also play a significant role in accidents. Rather, it is plausible to consider that advancements in car manufacturing technology and improved safety standards have contributed to the relatively lower frequency of modern cars being involved in such accidents.

It is important to acknowledge that the focus of this study was not to directly assess the impact of evolving technology and safety standards on accident rates. However, the observed pattern of older vehicles constituting a significant proportion of accidents in black spots implies that the enhanced safety features and design elements present in newer cars may have played a role in mitigating the occurrence of accidents in these specific areas. Further investigations into the specific mechanisms through which modern cars exhibit improved safety performance, particularly in black spots, would be valuable in confirming this hypothesis and informing future policy decisions in the field of traffic safety.

*6.4. Limitations*

Despite the valuable insights provided by the current research regarding the accidents transpiring in black spots, it is crucial to acknowledge certain limitations that may impact the generalizability and comprehensiveness of the findings. Recognizing these limitations can result in a more nuanced understanding of the research outcomes and the implications they hold.

It is essential to note that the study focused exclusively on the region of northern Greece and the specific time frame of 2014 to 2018. The findings, therefore, may not be representative of the broader traffic patterns and characteristics of other regions within Greece or different time periods. Variations in road infrastructure, and enforcement policies across regions and time frames could influence the results and limit their applicability beyond the study area.

Moreover the investigation primarily relied on quantitative data analysis. A method was designed to deal with the identification of black spots as a supervised learning problem in the context of machine learning, while this approach provided valuable statistical insights, it may not fully capture the complex interplay of various factors contributing to accidents, such as driver behaviour, environmental distractions or other temporal phenomena. Augmenting the dataset with in-depth interviews of drivers and stakeholders or perhaps gathering real-time accident data from the collided vehicles via Internet of Things (IoT) sensors, could have offered a more comprehensive understanding of the underlying causes of accidents in black spots.

**7. Conclusions**

In conclusion, road safety and black spot identification is a critical and ongoing concern for both public and private organizations. The identification of black spots, areas of higher risk for road accidents, is a spatiotemporal phenomenon that requires the integration of various data sources and advanced analytical techniques.

The present study involved compiling the Black Spot Dataset of North Greece (BSNG) by collecting data on road accidents and safety from various sources, including police reports, construction agencies, and academic experts. The data was organized and cleaned using spreadsheets, and features such as accident location, incident details, driver and vehicle information were extracted. Anonymization techniques were used to remove personal information from the data. The resulting dataset provided insight into the factors

contributing to accidents in specific locations and was used to identify black spots on roads in North Greece.

The study proposed a four-step approach that includes transforming each variable into labelled and one-hot encodings, using a self-supervised deep learning architecture to reduce input features, augmenting the feature vectors, and using a classifier to approximate a binary class problem. It provides a potential solution to the challenges associated with identifying black spots, such as limited data availability, data quality, and difficulties in accurately assessing factors contributing to road accidents.

It is important to note that black spot identification is a complex and dynamic field that requires continued research and improvement in methodologies. Nevertheless, the efforts made towards road safety and black spot identification will bring us one step closer to creating a safer and more efficient road network for all users. Overall, this study contributes to improving road safety by providing a publicly available dataset and a highly accurate black spot identification method.

**Author Contributions:** I.K., A.K., V.P., G.B., C.K., V.S. and G.P. contributed equally to this work. All authors have read and agreed to the published version of the manuscript.

**Funding:** This research received no external funding.

**Institutional Review Board Statement:** Not applicable.

**Informed Consent Statement:** Not applicable.

**Data Availability Statement:** Publicly available datasets were analysed in this study. This data can be found here: https://github.com/iokarama/BSNG-dataset (accessed on 15 June 2023).

**Acknowledgments:** Special thanks are given to the Hellenic Statistical Authority for providing the anonymized data related to road accidents that occurred during the years 2014–2018 on the road network of Macedonia and Thrace.

**Conflicts of Interest:** The authors declare no conflict of interest.

## Abbreviations

The following abbreviations are used in this manuscript:

| | |
|---|---|
| BSNG | Black Spot of North Greece |
| ELSTAT | Hellenic Statistical Authority |

## Appendix A

**Table A1.** Description of variable *Year*.

| Variable | Unit | Value | Correspondence |
|---|---|---|---|
| | | 1 | 2014 |
| | | 2 | 2015 |
| **Year** | year | 3 | 2016 |
| | | 4 | 2017 |
| | | 5 | 2018 |

**Table A2.** Description of variable *Month*.

| Variable | Unit | Value | Correspondence |
|---|---|---|---|
| | | 1 | January |
| | | 2 | February |
| | | 3 | March |
| | | 4 | April |
| | | 5 | May |
| | | 6 | June |
| **Month** | month | 7 | July |
| | | 8 | August |
| | | 9 | September |
| | | 10 | October |
| | | 11 | Nomvember |
| | | 12 | December |

**Table A3.** Description of variable *Weekday*.

| Variable | Unit | Value | Correspondence |
|---|---|---|---|
| **Weekday** | day | 1 | Monday |
| | | 2 | Tuesday |
| | | 3 | Wednesday |
| | | 4 | Thursday |
| | | 5 | Friday |
| | | 6 | Saturday |
| | | 7 | Sunday |

**Table A4.** Description of variable *Daylight*.

| Variable | Unit | Value | Correspondence |
|---|---|---|---|
| **Daylight** | hours | 0 | 19–05 |
| | | 1 | 06–18 |

**Table A5.** Description of variable *Traffic class*.

| Variable | Unit | Value | Correspondence |
|---|---|---|---|
| **Traffic class** | vehicle/hour | 1 | 0–1000 |
| | | 2 | 1001–2000 |
| | | 3 | 2001–3000 |
| | | 4 | 3001–5000 |
| | | 5 | 5001–8000 |
| | | 6 | >8001 |

**Table A6.** Description of variable *Roadway type*.

| Variable | Value | Correspondence |
|---|---|---|
| **Roadway type** | 1 | Tarmac |
| | 2 | Cement |
| | 3 | Gravel |
| | 4 | Cobbled |
| | 5 | Dirt |
| | 6 | Other |

**Table A7.** Description of variable *Atmospheric conditions*.

| Variable | Value | Correspondence |
|---|---|---|
| **Atmospheric conditions** | 1 | Good weather |
| | 2 | Strong winds |
| | 3 | Frost |
| | 4 | Fog |
| | 5 | Light rain |
| | 6 | Rain |
| | 7 | Gale |
| | 8 | Storm |
| | 9 | Hail |
| | 10 | Snow |
| | 11 | Smoke |
| | 12 | Dust |
| | 13 | Other |

**Table A8.** Description of variable *Roadside environment*.

| Variable | Value | Correspondence |
|----------|-------|----------------|
| **Roadside environment** | 1 | Habited |
|  | 2 | Inhabited |

**Table A9.** Description of variable *Road surface conditions*.

| Variable | Value | Correspondence |
|----------|-------|----------------|
| **Road surface conditions** | 1 | Normal |
|  | 2 | Wet |
|  | 3 | Slippery |
|  | 4 | Frozen |
|  | 5 | Snow |
|  | 6 | Other |

**Table A10.** Description of variable *Lane divider*.

| Variable | Value | Correspondence |
|----------|-------|----------------|
| **Lane divider** | 1 | Visible |
|  | 2 | Not visible |
|  | 3 | Not present |

**Table A11.** Description of variable *Road Narrowness*.

| Variable | Value | Correspondence |
|----------|-------|----------------|
| **Road Narrowness** | 1 | Normal |
|  | 2 | Narrowing |

**Table A12.** Description of variable *Lane direction sign*.

| Variable | Value | Correspondence |
|----------|-------|----------------|
| **Lane direction sign** | 1 | Visible |
|  | 2 | Not visible |
|  | 3 | Not present |

**Table A13.** Description of variable *Sequential turns*.

| Variable | Value | Correspondence |
|----------|-------|----------------|
| **Sequential turns** | 0 | False |
|  | 1 | True |

**Table A14.** Description of variable *Road gradient*.

| Variable | Value | Correspondence |
|----------|-------|----------------|
| **Road gradient** | 0 | Uphill |
|  | 1 | Downhill |
|  | 2 | Straight |

**Table A15.** Description of variable *Straightness*.

| Variable | Value | Correspondence |
|----------|-------|----------------|
| **Straightness** | 0 | False |
|  | 1 | True |

**Table A16.** Description of variable *Right turn*.

| Variable | Value | Correspondence |
|---|---|---|
| **Right turn** | 0 | False |
| | 1 | True |

**Table A17.** Description of variable *Left turn*.

| Variable | Value | Correspondence |
|---|---|---|
| **Left turn** | 0 | False |
| | 1 | True |

**Table A18.** Description of variable *Left barrier*.

| Variable | Value | Correspondence |
|---|---|---|
| **Left barrier** | 1 | Existent |
| | 2 | Non existent |

**Table A19.** Description of variable *Right barrier*.

| Variable | Value | Correspondence |
|---|---|---|
| **Right barrier** | 1 | Existent |
| | 2 | Non existent |

**Table A20.** Description of variable *Left edge line*.

| Variable | Value | Correspondence |
|---|---|---|
| **Left edge line** | 1 | Visible |
| | 2 | Not visible |
| | 3 | Not present |

**Table A21.** Description of variable *Right edge line*.

| Variable | Value | Correspondence |
|---|---|---|
| **Right edge line** | 1 | Visible |
| | 2 | Not visible |
| | 3 | Not present |

**Table A22.** Description of variable *Driver's age*.

| Variable | Value | Correspondence |
|---|---|---|
| **Driver's age** | 1 | <26 |
| | 2 | 26–64 |
| | 3 | >64 |

**Table A23.** Description of variable *Driver's gender*.

| Variable | Value | Correspondence |
|---|---|---|
| **Driver's gender** | 1 | Female |
| | 2 | Male |
| | 9 | Undefined |

**Table A24.** Description of variable *Accident severity*.

| Variable | Value | Correspondence |
|---|---|---|
| **Accident severity** | 0 | Unharmed |
| | 1 | Deceased |
| | 2 | Heavily wounded |
| | 3 | Wounded |

**Table A25.** Description of variable *Vehicle age*.

| Variable | Unit | Value | Correspondence |
|---|---|---|---|
| **Vehicle age** | year | −7 | <1 year |
| | | 1 | 1–2 |
| | | 2 | 3–5 |
| | | 3 | 6–10 |
| | | 4 | 11–15 |
| | | 5 | 16–19 |
| | | 6 | >20 |

**Table A26.** Description of variable *Mechanical inspection*.

| Variable | Value | Correspondence |
|---|---|---|
| **Mechanical inspection** | 1 | Yes |
| | 2 | No |
| | 3 | Not required |
| | 9 | Unknown |

**Table A27.** Description of variable *Vehicle type*.

| Variable | Value | Correspondence |
|---|---|---|
| **Vehicle type** | 1 | Private car |
| | 2 | Public car |
| | 3 | Military, police, armed forces, diplomat's car |
| | 4 | Company car |
| | 5 | Excavator |
| | 6 | Truck <3.5 t |
| | 7 | Truck >3.5 t |
| | 8 | Mobile home |
| | 9 | Tow truck |
| | 10 | Private bus |
| | 11 | Public transportation bus |
| | 12 | Public intercity bus |
| | 13 | School bus |
| | 14 | Sightseeing bus |
| | 15 | Military, police, armed forces, diplomat's bus |
| | 16 | Ambulance (with patient) |
| | 17 | Ambulance (without patient) |
| | 18 | Firetruck |
| | 19 | Trolley |
| | 20 | Tanker |
| | 21 | Bicycle |
| | 22 | Moped motorcycle <49 cc |
| | 23 | Motorcycle 50–115 cc |

**Table A27.** *Cont.*

| Variable | Value | Correspondence |
|---|---|---|
| | 24 | Motorcycle 116–269 cc |
| | 25 | Motorcycle 270–730 cc |
| | 26 | Motorcycle >730 cc |
| | 27 | Tricycle |
| | 28 | Agricultural tractors |
| | 29 | Other agricultural vehicles |
| | 30 | Train |
| | 31 | Tram |
| | 32 | Unknown |

## Notes

1. An example of data provided by ELSTAT (2014): https://www.statistics.gr/el/statistics/-/publication/SDT04/2014 (accessed on 15 June 2023).
2. Hellenic legislation on National guidelines regarding road safety checks: ΦΕΚ Β 1674, 13 June 2016.
3. ELSTAT is responsible for generating new statistical data driven by its commitment to fulfil obligations towards Eurostat, as well as various European and international organizations. These obligations encompass adhering to newly established European regulations, ensuring compliance with prevailing regulations, fulfilling questionnaire requirements, updating databases, and more.

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
