# Peer review of "Deep Learning-Based Black Spot Identification on Greek Road Networks"

_data, 2023_

Round 1

Reviewer 1 Report

Dear authors; many thanks for the oportunity to revise this work. The paper is well structured and it is the written is clear and I am grateful to them for this.

In general:

Road safety is an importan concern worldwide, and Europe leading of the good numbers in safety indicators as death by inhabitant, were obtained because the effort and work of research and gubernamental comunities together.

The authors present several methods used for black spot prediction, including traditional statistical analysis, machine learning, and deep learning methods, which are well described in different sections as

The overview of the different black spot prediction methods, their strengths and limitations, and to discuss their potential applications in the field of road safety. The contribution of this work is threefold as it offers an in depth literature review, it publishes a novel dataset on black spot identification (BSNG), and it establishes an identification benchmark of these problematic areas on Greek road networks.

They offer an interesting study, using sound methodology treebased

machine learning techniques including Decision tree (DT), Random Forest (RF),

Adaptive boosting (AdaBoost), and Extremely Randomized Trees (ExtraTrees), as well linear and non-linear SVM, within others, to deal with the identification of black spots, and finally select the best methodology to this end.

I have some concerns:

·       Up to the reviewr know, Europe have been done important work, for the armonization of the way and protocols to obtain the road accident data, in the countries belonging to UE. Is this the case of the Road Traffic Collision Reports (RTC) by ELSTAT?,

·       The data obtained, is shared with Eurostat, or  IRTAD?

·       It could be interesting for readers, to offer an overview on how the methodoly applied to predict road accidents and data used are related. The state in the paper, the methodologies vary in their level of complexity, data requirements, and accuracy, as well on the data used.

·       As the road accidents are under the influence of a great variety of factors (road design and infrastructure, driver behavior, and weather conditions), and the RTC database contain information of them, the authors can offer an analysis on the variables factors effects as the importance  provided by RF, o SVM.

·       In Table 2 the authors offer the statistics of variables of the helenic database, and I have at least two concerns: (in relation of variable type and class -levels of variable)

·       The databases contains diferent data type: continuous, categorical, ordinal, etc., and them can no be treated as iquals, the statistics depending of the variable type.

·       In Table 2, all of them are considered as continuos variables. It could be because they are re-codified as numeric as the code used for transformation. And the meaning can be different to those original.

·       The reviewer consider this issue has to be explained better.

·       It is clear “Black spot” is a cathegorical variable with two levels: 0 -1, the mean has not any sense

·       The authors have to describe the variables of the RTC database as well the levels or the meaning of the codification , for example:

·       The Interval [1,13] of the Atmospheric conditions variable, denote 13 levels or it denote a codification?, what mean each one?

·       Gender = 9 what does mean?

·       Driver age 1-3? means?

Reviewer 2 Report

Road safety is a hot topic in traffic control and management. Black spot identification is essential in traffic accident prediction and prevention. The authors used deep learning to identify the black spot of Greek road networks. It is exciting and valuable in traffic accident prevention. The paper is well-written and organized. It can be accepted in its current form.

Reviewer 3 Report

The study analyzes the effectiveness of different roadside black spot prediction methodologies, identifying their advantages and limitations. I consider that this is an interesting topic with important practical applications. In general, the paper is well presented, so I will only make some minor suggestions.

The introduction and the review of the literature are very complete, contextualizing the topic addressed as well as all the variables of interest of the study. The methodology is adequate and correctly presented.

The results are clear and adequately presented. However, I recommend further development of the discussion section. This section should not be a mere summary of the results but should explain and contrast the data obtained with other studies in this field of research. Thus, the authors should answer questions such as: were the results in accordance with expectations, are the results congruent with other similar research? And, if not, what elements explain the discrepancies that have occurred?

I also recommend including a specific section with the limitations of the study and future lines of research.
